# Is There a Risk of Misinterpretation of Potassium Concentration from Undetectable Hemolysis Using a POCT Blood Gas Analyzer in the Emergency Department?

**DOI:** 10.3390/medicina59010066

**Published:** 2022-12-28

**Authors:** Marianna Nigro, Gabriele Valli, Maria Luisa Marchionne, Dario Sattarinia, Fabiana Silvestrini, Daniele De Pietro, Simone Fazzini, Giorgia Roselli, Andrea Spallino, Valentina Praticò, Enrico Mirante, Ersilia Castaldo, Francesco Rocco Pugliese, Claudia Cicchini, Carlo Ancona, Francesca De Marco, Maria Pia Ruggieri, Salvatore Di Somma

**Affiliations:** 1Postgraduate School of Emergency Medicine, Sapienza University of Rome, 00185 Rome, Italy; 2Emergency Department, San Giovanni Addolorata Hospital, 00184 Rome, Italy; 3Emergency Department, S. Eugenio Hospital, 00144 Rome, Italy; 4Emergency Department, Sandro Pertini Hospital, 00157 Rome, Italy; 5GREAT Network Italia, 00191 Rome, Italy

**Keywords:** blood samples, pre-analytical errors, POCT arterial blood gasses analysis in ED, POCT device for hemolysis detection

## Abstract

*Background and Objectives*: Hemolysis is reported to be present in up to 10% of blood gas specimens in the central lab; however, few data on the incidence of hemolysis using a point-of-care testing (POCT) blood gas analysis are available in the setting of the emergency department. The aims of this study were: (1) to analyze the prevalence of hemolysis in blood gas samples collected in the ED using a POCT device; and (2) to evaluate the impact of hemolysis on blood sample results and its clinical consequences. *Materials and Methods*: We collected 525 consecutive POCT arterial blood gas samples using syringes with electrolyte-balanced heparin within 3 different EDs in the metropolitan area of Rome. Immediately after the collection, the blood samples were checked for the presence of hemolysis with a POCT instrument (i.e., HEMCHECK, H-10 ^®^). The samples were then subsequently processed for blood gasses, and an electrolytes analysis by a second operator blinded for the hemolysis results. A venous blood sample was simultaneously collected, analyzed for it’s potassium value, and used as a reference. *Results*: Of the samples, 472 were considered for the statistics, while 53 were excluded due to the high percentage of hemolysis due to operator fault in carrying out the measurement. The final mean hemolysis per operator was 12% (±13% SD), and the total final hemolysis was 14.4%.Potassium (K^+^) was significantly higher in the hemolyzed group compared with the non-hemolyzed sample (4.60 ± 0.11 vs. 3.99 ± 0.03 mEq/L; *p* < 0.001), and there were differences between arterial potassium versus venous potassium (D(a-v) K+, 0.29 ± 0.06 vs.−0.19 ± 0.02 mEq/L, *p* < 0.01). A Bland–Altman analysis confirmed that hemolysis significantly overestimated blood potassium level. *Conclusion*: Almost 12% of POCT blood gas analysis samples performed in the ED could be hemolyzed, and the presence of this hemolysis is not routinely detected. This could cause an error in the interpretation of the results, leading to the consideration of potassium concentrations being below the lower limit within the normal limits and also leading to the diagnosis of false hyperkalemia, which would have potential clinical consequences in therapeutic decision-making in the ED. The routine use of a POCT hemolysis detector could help prevent any misdiagnoses.

## 1. Introduction

Error in the medical field is usually referred to as “any preventable event that may cause or lead to inappropriate medication use or patient harm”, and prevention of such events must be the primary goal of the healthcare system [1]. An error event can lead to misdiagnosis and delayed or failed treatment, having great impact on healthcare performance and patient safety; it can occur due to malpractice and depend on human behavior, but it could also result from an error occurring in the processing of laboratory data. Laboratory errors can be difficult to identify without a specific test, and for this reason, the issue of laboratory errors is receiving increasing attention [1]. The process of laboratory data analysis consists of a pre-analytical phase, an analytical phase, and a post-analytical phase, and each of these steps are not free from the possibility of error [2]. Majority of errors occur in the extra-analytical phases of sample collection and storage. The frequency of error in this step accounts for 60–70% of the total errors that occur throughout the entire process [3].

Hemolysis represents the most evident error in the pre-analytical phase, as 40–70% of sample rejections are primarily due to the presence of hemolysis, followed by insufficient sample volume (10–20%), the use of an inappropriate container (5–15%), and the evidence of visible clots within the blood sample (5–10%) [3]. The term hemolysis refers to the breakage of red blood cells, resulting in the release of free hemoglobin into the plasma. After centrifugation of the blood sample, a process that helps to identify the phenomenon, a distinct pinkish discoloration of the serum may be noticed. This coloration is visible if the sample contains at least 0.5% of lysed erythrocytes and a free hemoglobin concentration greater than 0.3–0.5 g/L [1].

The emergency department (ED) is one setting of care where hemolysis phenomena are more frequent [1]. A survey promoted by the Working Group on Laboratory Errors and Patient Safety (WG-LEPS) of the International Federation of Clinical Chemistry and Laboratory Medicine (IFCC) [4] and the Global Preanalytical Scientific Committee (GPSC) [5], involving 391 clinical laboratories around the world, showed that approximately 53% of hemolyzed samples are ED samples [6]. This issue is determined by the working environment in the ED, which is often overcrowded and is a place where speed and efficiency are essential. While central laboratories systematically checked blood samples for hemolysis to limit this problem, the same does not happen for a point-of-care system (POCT) such as a blood gas analyzer [7]. Arterial blood gas analysis (BGA) is one of the most widely used technologies in the ED [8] based on the large amount of crucial information that it can provide in the critical area and, when used as a POCT system directly in the emergency room, reduces the waiting time for results and can improve decision-making and diagnostic processes [9]. Recently, this new technology was also found to be applicable in venous sampling analyses for metabolites and electrolytes detection (bilirubin, glucose, creatinine, urea, and Troponin I) [10]. However, the blood samples in POCT cases are not usually checked for hemolysis. This is critical if we consider that the parameters most affected by hemolysis are potassium, pH, PO_2_, and pCO_2_ [6].

However, hemolysis in arterial samples have not been studied in-depth, and even it’s frequency is not clear, having a range between 1.2% [11] and 7.9% [9]. Thus, given the limited data on this topic, we believe that our study may be of interest and bring consistent benefits to clinical practice.

The aims of this study were to: (1) estimate if the percentage of hemolysis in arterial samples collected using a POCT system accessible in ED in real life might be much higher than 8%; and (2) evaluate the hemolysis-induced error in the dosage of some values (i.e., potassium) and the chance of its relevant clinical impact.

## 2. Materials and Methods

### 2.1. Population

The study was conducted in the ED of three different hospital sites (San Giovanni Addolorata Hospital, Sant’Eugenio Hospital, Sandro Pertini Hospital) in the city of Rome from the 17th of March 2021 to the 30th of April 2022. All patients in whom BGA was used for analysis were deemed eligible for our study, regardless of the demographics, comorbidities, or presented complaints. There was no numerical limit on the number of BGAs collected. We followed the principles and recommendation of the World Medical Association (WMA) Declaration of Helsinki, and we collected data according to good clinical practice. The study was approved by the competent Ethical Commit Lazio 2 as an addendum of the study VAG-COVID19 (letter n 4D7ED12B-72A9-65F3-27B3-CA30D5F2CE31@telecompost.it, 17/03/2021; protocol approval number 0157973/2020, 30/09/2020).

### 2.2. Study Design

We collected blood samples from every patient in the EDs requiring rapid arterial blood analysis. Patients in cardiac arrest were not included in the sample analyzed. This procedure was left up to the nursing staff. The arterial samples were collected from the radial artery site in syringes with electrolyte-balanced heparin (BD Preset^™^Eclipse^™^Arterial Blood Collection Syringes, Becton, Dickinson and Company, Belliver Industrial Estate, Belliver Way, Roborough, Plymounth, PL6 7BP, UK), intended for blood gas analysis on a RAPIDPoint^®^ 500e blood gas analyzer (Siemens HealthCare, Henkestr. 127, 91052 Erlangen, Germany). The details of the syringe features were a volume of 3.0 mL, 80/50 units of heparin (IU per syringe/per mL of blood), recommended mean fill volume of 1.6 mL, 0.6-gauge needle, and needle length of 25mm. A venous blood sample was collected with the arterial sample to dose the venous potassium as a reference value for potassium. Immediately after the sampling was performed, as part of routine care, an ID number was assigned to the sample, and all of the patients’ identifiers were removed from the syringe. A second operator, a medical resident, checked the blood for hemolysis with the point-of-care (POCT) Hemcheck (H10) ^®^ system (Hemheck Sweden Company, Solona, Sweden). Every member of this study received online training for the correct use and interpretation of the device results with the team of Hemcheck Sweden Company before the beginning of the study. The sample was included in the study if the amount of blood in the syringe was considered sufficient for performing all analyses needed for the study. The only exclusion criterion was an insufficient amount of blood in the sample, which was set to less than 300 μL. The Hemcheck (H10) ^®^ system analyzed approximately 100 μL of blood. The POCT system quantitatively and photometrically analyzed the free hemoglobin in the plasma as already previously described [9]. The user can define which values should be considered positive. In this study, the limit of 50 mg/dL was considered positive for hemolysis. The cutoff was adjustable using software settings in the Helge H10 s-system.

Concordance between the Hemcheck H10’s capacity to detect hemolysis has been validated in several previous papers [9]. In their paper, Duhalde et al. tested the capacity of H10 to detect hemolysis in over 1270 blood samples and compared the results with the central laboratory results performed on an AU680 instrument (Beckman Coulter, Brea, California, USA). A direct correlation was found between the two methods, with an R2 value of 0.91 (*p* < 0.05), confirmed by a Spearman’s correlation test for non-normal distributed data, which showed a correlation coefficient of 0.85 (*p* < 0.0001). The mean difference between the methods was 7.5 (SD 21.3) mg/dL, and the limits of agreement were 49.1 and 34.2 mg/dL.

After the rapid analysis with the Hemcheck (H10) ^®^ device, the sample was analyzed by another operator, a qualified nurse who was blinded to the results of the hemolysis test, and they recorded the AGB results. Data on the Hemcheck (H10) ^®^ results and clinical and laboratory results were separately collected and recorded by the two operators on an online database to ensure that the results were blinded to the investigators and clinicians.

### 2.3. Data Collection

An online questionnaire was designed, and the residents had to collect the following data (Table 1):

### 2.4. Statistical Analysis

The statistical significance was determined with a threshold of 95% (α < 0,05). The whole statistical examination was performed using the statistical software StatPlus (StatPlus Pro v7©, AlaystSoft Inc., Walnut, CA, USA).

We defined the sample size according to our historical registry used in a previous paper on the cost analysis of the emergency department [12]. We extrapolated data on the BGA performed in ED, showing that there was at least 3–4 analyses per day (almost 3500 BGA in the three different hospitals involved); we assumed an 8% of prevalence of Hemolysis, an error of 2%, and IC of 95%. From these data, we calculated a sample size of 588 to obtain a representative sample, and at least 50 samples were hemolyzed. 

With the aim of minimizing possible errors due to the operator, the percentage of hemolysis of each operator dedicated to the measurement of hemolysis was analyzed. In this case, we defined the operator as the person who executed the blood drawing. The operators who had a percentage level of hemolyzed samples greater than twice the standard deviation were excluded from the analysis (Figure 1 and Table 2). The remaining sample was then divided in two groups, non-hemolyzedand hemolyzed, for comparison.

Continuous variables were shown as the mean ± SD. The differences between means of the different groups were tested using the t-Student test. The categorical variables were summarized in a crosstab, expressed as the percentage of the group, and analyzed using an χ2 test, and if the test obtained a significant result, it was further analyzed using a z- test. The correlations between variables were analyzed with a Pearson regression analysis and verified with a univariate linear regression analysis(Shapiro–Wilk test). The coefficients of the linear regression were estimated with the method of minimal squares, and the strength of the correlation was expressed using the Pearson coefficient (R). A Bland–Altman plot [13] was utilized to highlight the differences between the level of arterial potassium and venous potassium. The average error of the measurement of this electrolyte in the two groups are shown in dedicated plots and summarized as the mean differences ± 95% of the confidence interval.

## 3. Results

A total of 525 arterial samples were collected, of which an initial statistical analysis was carried out, starting from the percentage of hemolysis of each individual operator in the study (Figure 1).

The mean hemolysis was 18% (±19% SD), but three operators (n° 5, n° 8, and n° 13) reported a percentage of hemolysis that was +2SD out of the means, and they were thus eliminated. The final group included 472 arterial samples and hemolysis, defined as those with more than 0.5 g/L of free hemoglobin in plasma, was present in 12% (±13% SD) of all samples tested (Table 2). No differences between the non-hemolyzed and hemolyzed samples were found on the distribution of the main symptoms that led patients to the emergency visit (Table 3).

Table 3 classifies the sample according to the symptom that led the physician to perform the BGA and shows the result of the analysis of BGA, with the electrolytes and metabolites divided into two groups (hemolyzed and non-hemolyzed samples). Of the six values analyzed, in the hemolyzed samples compared with the non-hemolyzed samples, we found a statistically significant increase in potassium by 15.29% (+0.61 mEq/L; *p*-value < 0.001). So, the hemolyzed samples have higher potassium levels, both as an absolute value in the arterial blood samples and as a difference between the arterial and venous samples (Δ(a-v) K^+^) of the same patient in a contextual sampling. The pCO_2_ was also increased in the hemolyzed samples (+ 4.20%) as well as the lactate (+26.3%), but these were not statistically significant. The pO_2_ value was non-statistically significantly decreased in the hemolyzed samples (−8,6%). The pH and Ca^2+^ values remained unchanged in the two groups under analysis. The concordance between the two measures (BGA and venous) was also evaluated (Figure 2), and we found a strong correlation in potassium dosage between arterial and venous sample (R^2^ 0.89 in non-hemolyzed, *p* < 0.001 and R^2^ 0.74 in hemolyzed, *p* < 0.001, respectively). However, the Bland–Altman analysis (Figure 3, plot A and plot B) shows that in the case of the non-hemolyzed group (plot A), the arterial values are almost identical to the values measured in the venous (Δ_(a-v_) K^+^: −0.07, 95% CI (−0.58,0.38)), with an average bias of about 0.5 mEq/L. Meanwhile, in the hemolyzed group (plot B), the mean of the differences is frankly above zero (Δ_(a-v_) K^+^: 0.29, 95%CI (−0.70,128)), with an average bias of about 1 mEq/L.

In Figure 4, the potassium values were categorized in five different intervals according to the values of the venous sample: hypokalemia (<3.5 mEq/L),lower normal range (3.5 to 4.1 mEq/L), median range (4.1 to 4.7 mEq/L), upper normal range (4.7 to 5.3 mEq/L), and hyperkalemia (>5.3 mEq/L); the error bars represent 95% confidence intervals of the hemolyzed (full black circles) and non-hemolyzed (open white circles) groups. As shown in the figure, in the hemolyzed group, the mean error in the potassium level measured on BGA of the <3.5 group falls into the diagnosis of normokalaemia, which does not occur in the non-hemolyzed group. On the other hand, for samples whose values are between 4.1 and 5.3 mEq/L, even if the mean of the hemolyzed samples is within the range of normokalaemia, the upper limit of the systematic error (+1.28) could lead to a misdiagnosis of hyperkalemia.

More than 77% of hemolyzed samples have a higher value of K+ in arterial blood than in venous blood. The 15.2% of the K+ values change from venous hypokalemia to artetial normokalemia, while there was only a 5.08% shift from venous normokalemia to arterial hyperkalemia. The same analysis in the non-Hemolyzed samples does not show any change.

Finally, Figure 5 shows the distribution of ranges of hypokalemia, normokalemia and hyperkalemia between the hemolyzed and non-hemolyzed samples. 

## 4. Discussion

The main finding of our study is that the average rate of POCT BGA hemolysis in the ED is higher than expected and that it could lead to the misdiagnosis of electrolyte imbalance [9,14].

The normal concentration of free hemoglobin is usually around 0.22–0.25 g/L in the serum and between 0.10 and 0.13 g/L in the plasma. However, consensus has established that the threshold for effectively defining a hemolyzed sample is the presence of 0.5 g/L of free hemoglobin in the specimen [3]. In fact, some authors define the cut-off of 0.5g/L free hemoglobin in the sample as the limit above which it can cause interference in sample analysis. Many studies also suggests that it is better to avoid processing hemolyzed samples [15].

Based on the mechanism that determines the phenomenon, a distinction can be made between in vivoand in vitro hemolysis. In the first case (in vivo), hemolysis is attributable to the patient’s clinical conditions (autoimmune diseases, mechanical heart valves, adverse reaction to transfusion, etc.) and typically accounts for less than 2% of the hemolysis found. Meanwhile, the in vitro hemolysis is mainly caused by inadequate collection or improper transport and storage of specimens before the analysis. Both kinds of hemolysis produce the same type of interference in the sample analysis [1]. In vitro hemolysis could be a clinical problem, as it interferes with the reliability of the results provided to physicians and is unrelated to the patient’s clinical condition. This result is an error that risks endangering the patient’s health as well as their discomfort and an increase in costs due to the need to repeat the blood sample [16]. The analytes most affected by hemolysis due to the loss of red blood cell content in plasma are potassium, LDH, and AST. Some authors also describe the interference of hemolysis in troponin analysis [17].Potassium is the most relevant indicator that an incorrect dosage of this electrolyte could have for clinical implications [16]. 

Our results show a hemolysis rate of 12% among the arterial samples analyzed. This percentage is higher than the percentage described in the literature, as we can see from the studies conducted by Duhalde et al. in 2019, where a hemolysis rate of 7.9% was found in a total sample of 1270 specimens analyzed in laboratory [9], and by Salvagno et al., which found that in a previous study, only 4% of specimens were hemolyzed in a sample of 487 arterial BGA [18]. Similarly, Wilson et al. compared the occurrence of hemolysis in arterial samples from the ED and ICU; they found that out of 100 samples, the percentages of hemolysis in the ED and ICU samples were 6% and 3%, respectively [14].

The same authors discussed that the reason for these differences could be on account of factors that can cause in vitro hemolysis due to mechanical factors, such as an inappropriate needle size, excess blood flow rate, presence of arterial lines, or the need to perform an arterial puncture. They also considered the difference between the expertise of staff in the ICU and ED [14].

This study therefore confirms the need to accurately identify hemolyzed arterial samples, as even the Clinical and Laboratory Standards Institute C46-A2 guidelines suggest that arterial samples with hemolysis should not be analyzed [19].

The hemolysis of arterial samples can affect patients in a major way, especially in the ED. It can result in prolonging hospitalization due to the need to repeat the sampling, which confuses the diagnostic process with unreliable data. Furthermore, it creates a false sense of security in the clinician who does not know whether the sample is hemolyzed, as POCTs do not assess the presence of hemolysis [20]. In addition, hemolysis can significantly interfere with several analytes, in particular altering potassium values with the risk of increasing potassium concentrations, leading to the diagnosis of pseudo-hyperkalemia or mimicking normal potassium values and thus masking hypokalemia [21].

We found a statistically significant increase in potassium levels by +16% (+0.61 mEq/L; *p*-value < 0.001) in the hemolyzed samples compared withthe non-hemolyzed samples. Thus, the hemolyzed samples tended to have higher potassium values, both as an absolute value and as a difference (Δ(a-v) K) of the measured value of venous and arterial potassium of the same patient, in a contextual sampling. Our data confirm the possibility of incorrect therapeutic intervention (or non-intervention) if the clinician is unaware of the presence of hemolysis in the arterial sample analyzed.

Potassium disorders are common in patients attending EDs. Hypokalemia and hyperkalemia are very common electrolyte disorders, and if untreated, they can lead to potentially fatal cardiac conduction disturbances and neuromuscular disorders [22]. Pseudo-hyperkalemia can result in iatrogenic hypokalemia. In fact, the incorrect administration of therapy can lead to death through ventricular fibrillation/tachycardia, paralytic ileus, and respiratory depression [23]. Similar factors can cause masked hypokalemia, i.e., potassium being within normal ranges despite the patient being hypokalemic [24]. Unrecognized hypokalemia can also become potentially fatal for the patient, especially in the population group taking drugs that interfere with potassium homeostasis [25], who are administered sodium bicarbonate as a therapy for metabolic acidosis [26]; however, it is also a factor frequently associated with a worsening outcome, even in patients that are initially at low risk of deterioration [27]. Furthermore, our results seem to show that hemolysis can lead in some cases to a missed or incorrect diagnosis. As shown in Figure 4, when the potassium values were below normal, the average of the hemolyzed samples fell into the range of normokalaemia. In contrast, for samples at the high limits, the upper limit of the systematic error fell into the range of hyperkalemia.

From our sample, we eliminated the samples collected from three operators. This was because the personal hemolysis rate of each of them exceeded the mean of the total sample by +2SD. This finding may be associated with the fact that the correct use of POCT for the detection of hemolysis is associated with a large proportion of operator-related error, a major limit of this instrument.

## 5. Conclusions

Rapid detection of hemolyzed samples from a POCT blood gas analyzer in the ED is useful and should be an integral part of the diagnostic process to avoid possible misdiagnosis. The most impacted laboratory result from hemolysis is potassium levels. With the use of a POCT for detection of hemolyzed BGA samples in the ED, it is possible to not underestimate the presence of hypokalemia and hyperkalemia and avoid incorrect therapeutic choices due to incorrect data.

It should be taken into account the possible operator-related error that might interfere with the sample analysis.

Further multicentric studies are needed to better understand how much hemolysis interferes with the whole analysis of an arterial sample in the ED.

## Figures and Tables

**Figure 1 medicina-59-00066-f001:**
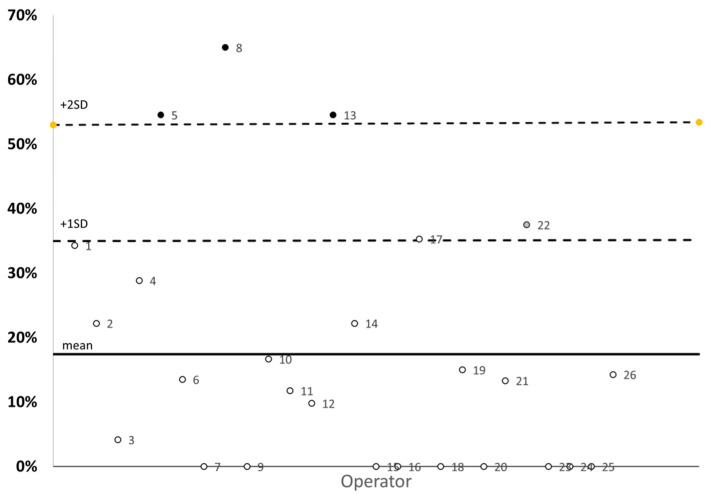
Percentage of hemolysis of each operator in the study. **○,** operators included in the statistical analysis, ●, operators excluded in the statistical analysis.

**Figure 2 medicina-59-00066-f002:**
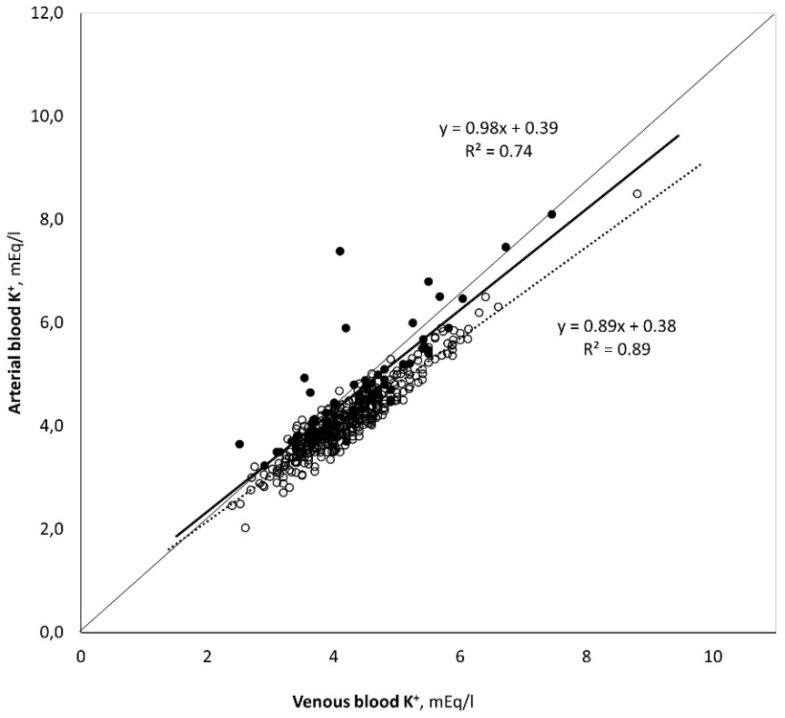
Measure of the concordance between the potassium value measured on the venous and arterial sample. (○ = non-hemolyzed; ● hemolyzed).

**Figure 3 medicina-59-00066-f003:**
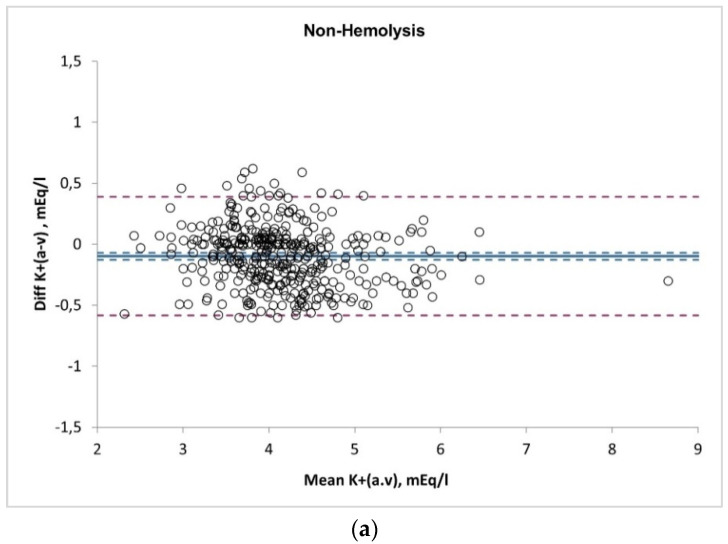
Bland–Altman Analysis. Plot (**a**) non-hemolyzed samples; plot (**b**) hemolyzed samples.

**Figure 4 medicina-59-00066-f004:**
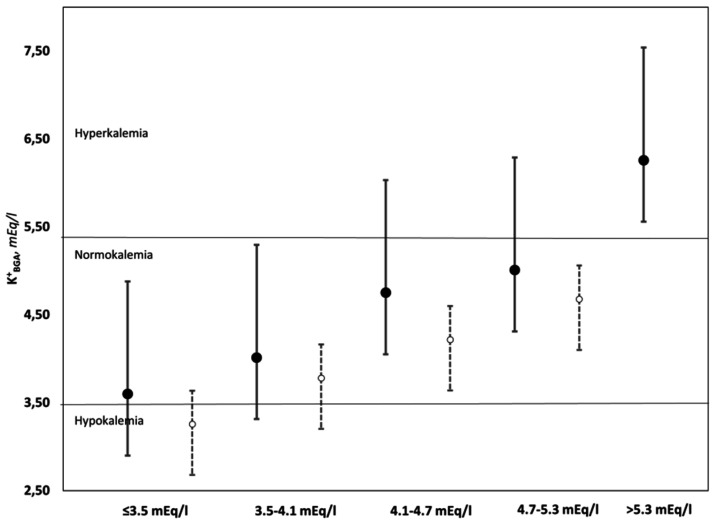
Comparison between the means and respective systematic errors of the hemolyzed (●) and non-hemolyzed (**○**) samples, divided into five ranges of values and three clinical conditions.

**Figure 5 medicina-59-00066-f005:**
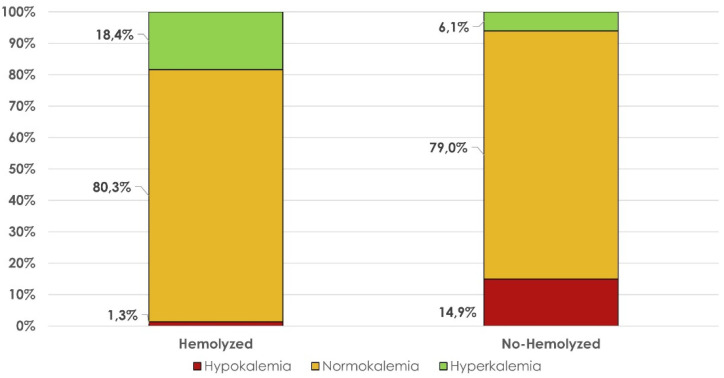
Distribution of hypo-, normo-, and hyperkalemia among hemolyzed and non-hemolyzed samples.

**Table 1 medicina-59-00066-t001:** Online questionnaire items to collect data. ED, Emergency Department; pCO_2_, partial pressure of carbon dioxide; pO_2_, oxygen partial pressure; BGA, blood gas analisis.

Online Questionnaire Items
Resident Doctor and responsible physician
ED’s Hospital site
Date and time of blood sample
Sex and date of birth
Clinical symptom on the Triage
Ml of blood collected and exclusions of criteria base on the quality and quantity of the sample
Method (arterial puncture or cannulation and device of syringe)
Evidence of hemolysis from the H10 analysis (yes/no)
Data from rapid arterial blood gas analysis and normal venous analysis: potassium, calcium, pH, pCO_2_, pO_2_, lactates from BGA potassium from venous sample

**Table 2 medicina-59-00066-t002:** Final Group. Operators 5, 8, and 13 showed a percentage of hemolysins of 55%, 65% and 55%, respectively, which was much higher than the average percentage of hemolysis.

	*n*	% Hemolysis	
Complete	525	18% mean	±19% SD
Operator 5	11	55%	
Operator 8	20	65%	
Operator 13	22	55%	
Final Group without operator out of +2 SD	472	12% mean	±13% SD

**Table 3 medicina-59-00066-t003:** Classification of the analyzed sample and the results, shown as the mean ± SE. The significance of the difference has been assessed by a t-test. Ca^++^, ionized calcium (mEq/L); K^+^ BGA, arterial potassium (mEq/L); La- (lactid acid (mmol/L)); pCO_2_, partial pressure of carbon dioxide (mmHg); pO_2_, oxygen partial pressure (mmHg); D(a-v)K, difference between arterial and venous potassium value.

	Non-Hemolysis	Hemolysis	
*n*	396	76	
Female, %	43%	38%	n.s.
Symptoms:			
Dyspnea	54.7%	64.5%	n.s.
Coma	9.2%	6.6%	n.s.
Chest Pain	8.4%	10.5%	n.s.
Abdominal Pain	7.1%	5.3%	n.s.
Fever	6.6%	9.2%	n.s.
Syncope	4.3%	1.3%	n.s.
Trauma	2.5%	2.6%	n.s.
Others	6.0%	-	*0.01*
K^+^_BGA_,mEq/L	3.99 ± 0.03	4.6 ± 0.11	<0.001
Ca^++^_BGA_, mEq/L	1.149 ± 0.003	1.149 ± 0.01	n.s.
pH	7.43 ± 0.01	7.42 ± 0.01	n.s.
pO_2_, mmHg	86.1 ± 1.7	78.7 ± 2.2	n.s.
pCO_2_, mmHg	38.1 ± 0.5	39.7 ± 1.4	n.s.
La-, mmol/L	1.9 ± 0.2	2.4 ± 0.4	n.s.
D_(a-v)_K^+^, mEq/L	−0.19 ± 0.02	0.29 ± 0.06	<0.01

## Data Availability

The data presented in this study are available on request from the corresponding author. The data are not publicly available due to patient confidentiality.

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
