# Peer review of "Is There a Risk of Misinterpretation of Potassium Concentration from Undetectable Hemolysis Using a POCT Blood Gas Analyzer in the Emergency Department?"

_medicina, 2022, doi:10.3390/medicina59010066_

Round 1

Reviewer 1 Report

General comment

Throughout the manuscript, use mmol/L – as recommended by many journals and medical societies- instead of mEq/L when referring to the K and Ca concentrations.

Also, once Emergency Department was firstly abbreviated as ED, use the abbreviation throughout the following text.

The manuscript do not adequately describe the concordance between a new instrument to detect hemolysis in whole blood with the detection in the central laboratory. This is a result of main interest for the readers.

Title

POCT is not (usually) a standard abbreviation; it should be described.

Abstract

The authors mention the lack of data on hemolysis incidence by POCT in the ED; however, such incidence can be found at least in references 7 and 12.

Briefly mention how the study samples were obtained (syringes, tubes, from lines, puncture, etc.).

In the results section, all samples sum 527, while in the material & methods 525.

The percentage (12%) or the number (76) of hemolyzed samples do not agree between them (76 hemolyzed samples account for 14.48% of 525 samples or for the 16.10% of 472 samples, while a 12% of hemolysis in 472 samples equals 56.64 hemolyzed samples or 63 if 525 are considered).

At the end of the Abstract., it should be mentioned that hemolysis could also cause false hyperkalemia.

Introduction

First paragraph

Indicate that the 60-70% of errors in the extra-analytical phase refer to total of errors, not to the number of samples reaching a clinical laboratory.

2nd paragraph

The authors should soften the affirmation that the ED is the site with the highest percentage of hemolyzed samples. Sometimes dialysis clinics or neonatal and adult ICUs have a similar or even higher percentage of hemolyzed samples. The percentage depends of the method used to detect it (visual hemolytic index) and of a fact that is often forgotten in many studies, as ICUs mostly use only whole blood for blood gases and electrolytes, whereas EDs do the same procedures, but more frequently send a simultaneous venous sample to central lab for further analyses where hemolysis is detectable in all drawn samples. This fact, as the authors suggest in their work, could underestimate % hemolysis in those clinical settings that mostly use POCT and whole blood for the main bulk of their analytical requests.

Indicate the organization pertains the Working Group on Laboratory Errors and Patient Safety.

“Arterial blood gas analysis (BGA) is one of the most…”; add to the text that not only arterial, but also venous blood it is frequently used for electrolytes and metabolites (bilirubin, glucose, creatinine, urea) measurement in the POCT instruments providing such analyses.

“However, the arterial blood gas samples are not usually checked.” I suggest refining the sentence, since the same happens for whole venous blood assessed in the POCT instruments.

The last sentence of the paragraph. As mentioned in the comments to the Abstract, hemolysis in POCT samples in ED settings have been previously analysed in other papers. However, due to the existing little evidence, data from current work will be useful for clinical practice.

Material and methods

As previously asked, more details on blood sampling (when syringes or tubes, from lines or by puncture, operator’s experience and differences between operators, etc.) must be provided. This is a critical step to put into context the observed % of hemolysis, as well as the differences among operators.

The same applies to POCT analyses. Who analysed the samples? What was their previous formation, not only in the BGA POCT analyzer, but also in the H-10 instrument?

Regarding operators excluded by high percentage of hemolyzed samples: Could the clinical status of the patients (dehydration, collapse, shock, severe hypotension) be related to the high percentage of hemolysis observed in these operators? In such case, operators are not guilty and the results from their drawings should be included in the calculations.

Regarding operators with “0” hemolyzed samples: Did they obtain a significant number of samples? Holding a “0” record in an ED after sampling many patients is quite difficult (in my own experience).

Statistics. Was the Gaussian distribution of variables assessed with parametric tests (e.g., t-test) checked before applying such tests?

Correct “Pearson”, “squares.”

Something lacks in the sentence “The average error of the measurement of this electrolyte in the two groups and ????”

Results

Samples included. The text refers to 473 samples, a newly appearing number in the text. Clarify.

‘Hemolysis was present in 11% (±12% SD) of all samples tested.’ Again, different numbers than the previously presented. The 12% of hemolyzed samples concurs with the Abstract, but the Table 1 set the SD at 13%. Review the data along the whole text.

Table 2 data. Symptoms in both columns sum just 100.0%; so, it should be assumed that only the main symptom of each patient was included in the Table. However, since ‘monosymptomatic patients’ are not frequent in the EDs, it is recommended to clarify the issue of symptoms classification in the Table heading (if required).

Table 2 results. Hemolyzed samples have, not “tend to have”, a higher K. If a pCO2 increase in the hemolyzed samples of +4.2% merited to be mentioned in the text, what about pO2 that was lower (-8.6%) and lactate that was higher (+26.3%)? The table lacks the p value for lactate comparison between hemolyzed and non-hemolyzed samples. Moreover, what was the concordance between the H-10 and the central lab in the hemolysis detection?

Figure 2. Use points instead of commas for the decimals in the equations.

Abbreviate the confidence interval as CI, not as IC.

Figure 4, in the text. What is “the waiting 95% confidence intervals of bias...? Otherwise, the figure is fully representative of the troubles that hemolysis could cause on the true K values. However, a new table showing how many K values changed from hypoK to normoK or from normoK to hyperK would be of great utility to highlight the frequency of the false classifications. The authors are encouraged to present such data; perhaps, as a flow-chart describing the whole sampling process (inclusion, ruling-out, true results in venous sampling, discrepancies from the truth in the POCT, usefulness of the hemolysis detection by the POCT tested).

Discussion

It is fully concordant with the Results and adequate. Only a few points.

1st paragraph.

‘Hemolysis in the ED is higher than expected’. To what data are the currently presented compared?

2nd paragraph.

There is no consensus on the free Hb (fHb) to be considered as significant hemolysis, as fHb interference depends on the biomarker evaluated biomarker and the methodology used for its measurement. Some biomarkers are unaffected by hemolysis, others by a fHb >1.0 g/L. This point should be included in the Discussion.

3rd paragraph.

Add to the paragraph that ‘in vivo’ hemolysis causes the same interferences as ‘in vitro’.

K not only increases in the circulation by red blood cells break; other cells (leukocytes, platelets, even tissue cells) contribute to increase K concentrations when broken.

Among the interfered biomarkers, cardiac troponin T and I should be mentioned.

4th paragraph.

What is said in the Abstract and the Introduction should concur with what is discussed here regarding the lack of data in the ED hemolysis detection with POCT methods.

In a future, new version of the Discussion, the concordance between H-10 and the central lab to detect hemolysis should be referred and discussed. It is of great interest to the readers.

Reviewer 2 Report

Thank you for the opportunity to review this ms. on the rate of hemolysis in arterial blood gas samples. I was very surprised to learn that in the three sites who contributed to this ms. have a really high rate of hemolysis in arterial blood glas analysis. 

Please allow me to suggest following comments to further improve this ms.:

1) In laboratory medicine, different cut-offs are used for different parameters such as potassium, troponin etc. I would suggest that the authors should further explain (and refer to related ms.), why they have used 0.5mg/L as a cutoff for hemolysis. 

2) The authors find that aBGA sampling was significantly higher in three operators compared to the other operators. Those samples were excluded for further analysis. I strongly believe that this cannot be done from the scientific point of view but indicates technical issues during blood sampling at those institutions. In addition, a rate of 12% of hemolysis is considerably higher than in references reported by these authors. Although I accept the arguments of the authors that hemolysis is relevant for proper management of patients, it is difficult to present data which may be biased due to inadquate techniques during blood sampling. The authors should address these issues adequately and should possibly adapt cutoff values for hemolysis. 

3) Please allow me to recommend that the introduction section and the discussion section could be considerably shortened. I would prefer to more focus on the main aims of this ms.

4) Authors should report a defined primary aim of this study and report a sample size calculation with power calculation. In addition, 

Minor comments:

1) Please use "open symbols" for non-hemolysis samples in Fig 3a (as used in fig 2) for congruent presentation of results.

2) Please use comparable symbols in Fig 4 for consistency.

Reviewer 3 Report

Dear authors,

Thank you for your interesting manuscript. I had a question while reading. I'll list them below.

1. Do emergency patients include cardiopulmonary arrest patients?

2. Are all the needles used the same size? If they are the same, how thick are they?

3. Are all arteries drawn from the same site (e.g. femoral artery or radial artery)? Also, did the results differ depending on which part of the artery the blood was drawn from?

4. The horizontal axis in Figure 1 has no units.

5.Although hemolysis is known to increase potassium, high potassium does not necessarily mean hemolysis. Is it possible to graph this sample for 1) high potassium without hemolysis 2) high potassium with hemolysis 3) normal or low potassium without hemolysis and normal or low potassium with hemolysis?

6. 

Round 2

Reviewer 3 Report

Dear Authors,

I think you have answered my question properly. In addition, it became very easy to understand by inserting "Fig. 5".

I recommend that your manuscript be published in "Medicina". Thank you for your good work.